# A Structural Time Series Analysis of the Effect of Quantitative Easing on Stock Prices

George B. Tawadros [1,] and Imad A. Moosa [2]

[1] Department of Economics, College of Business, Winona State University, Winona, MN 55987, USA
[2] Department of Economics, Kuwait University, Shaddadiya 46410, Kuwait
[*] Correspondence: george.tawadros@winona.edu

**Abstract:** In this paper, a structural time series model is estimated to analyse the effect of quantitative easing (QE) on stock prices for the US, UK and Japan. The model is estimated by maximum likelihood in a time-varying parametric framework, using the DJIA, S&P500, NASDAQ, FTSE100 and the NIKKEI225 as the dependent variable and the balance sheet of the respective Central Bank as an explanatory variable, along with the unobserved components that account for the behaviour of other explanatory variables that are not explicitly specified in the model. The results show that QE had a significant but not exclusive effect on the DJIA, S&P500 and the NASDAQ, suggesting that these stock prices are also affected by other missing variables and cyclical movements. However, for the UK and Japan, no effect of QE on the FTSE100 and the NIKKEI225 is found, suggesting that variables other than QE are important for the rise in these stock prices. One plausible explanation for this result is that perhaps QE becomes effective only after a certain threshold level is met.

**Keywords:** central banks; quantitative easing; structural time series modelling; stock prices

**JEL Classification:** E44; E51; G10

## 1. Introduction

Monetary history has long shown many examples where changes in monetary policy have led to significant changes in stock prices. For instance, the seminal work by Friedman and Schwartz (1971) highlights many examples for the US over the period 1867–1960, where an expansion in the money supply has led to significantly high stock valuations and faster economic growth. They also show instances where contractionary monetary policy has led to sharp falls in stock prices and recessions or depressions, such as what happened in the 1930s.

One type of monetary policy tool that has gained in popularity in recent times is that of 'quantitative easing' (QE). This requires the central bank to print new money in order to buy securities of different kinds and maturities from financial institutions. In so doing, security prices rise while yields fall, keeping interest rates low across a range of asset maturities. As Joyce et al. (2011a, 2011b, 2012), Joyce and Tong (2012), Fischbacher et al. (2013) and Galí and Gambetti (2015) note, the objective is to boost the real side of the economy because lower interest rates allow firms and households to spend more than they otherwise would, boosting economic activity (See, for instance, Palley (2011) and Szczerbowicz (2015)). Bernanke and Reinhart (2004) show that QE is used to reduce the cost of borrowing to a lower level than what would be achieved by using conventional open market operations, which targets a particular short-term interest rate.

Since the adoption of QE by Japan in 2001, debate has ensued about its effectiveness. Proponents of QE suggest that it has kept borrowing costs low for firms and households, created jobs, and saved economies from entering severe recessions or depressions. Those who oppose QE, on the other hand, contend that it is highly inflationary. As Blinder

(2013) and Galí and Gambetti (2015) show, inflation has only appeared in asset prices. For instance, because of the Subprime Crisis, the Dow Jones Industrial Average (DJIA) and the S&P500 in the US, along with the Nikkei225 in Japan, fell by more than 30 percent between September 2008 and January 2009. Over the same period, FTSE100 in Britain fell by more than 20 percent. By the middle of 2009, however, the stock markets of all three countries had reached their lowest point and had begun to recover because of the use of QE in response to the Subprime Crisis. By the end of October 2011, the DJIA and the S&P500 had increased by more than 50 percent, while the FTSE100 rose by 29 percent., but the Nikkei225 rose by only 12 percent. Joyce et al. (2012) and Joyce and Tong (2012) suggest that QE may be designed to significantly raise asset prices so that the wealth effect boosts the economy. In particular, when asset prices significantly increase, people and businesses feel wealthier, so that they are more likely to engage in greater economic activity. Additionally, as household and corporate balance sheets improve, these sectors are in a better position to engage in greater economic activity (Blinder 2013).

Since the end of the Subprime Crisis, the Federal Reserve of the United States, Bank of England, and the Bank of Japan, have all implemented successive rounds of QE, with varying degrees of duration and intensity. Stock prices have since risen sharply, even in the presence of rising interest rates. However, while conventional wisdom and stylized facts show that stock prices have risen significantly during the successive rounds of QE, there still is no consensus on whether QE has been the dominant catalyst causing the sharp rise in stock prices. Many commentators and practitioners have noted that the post-crisis rally cannot be solely attributed to the use of QE and that the bearish market of 2015 cannot be solely attributed to the end of QE.

The objective of this paper is to provide empirical evidence on the effects of QE implemented by the Federal Reserve of the United States, Bank of England, and the Bank of Japan, on the stock prices of these countries. Unlike the extant literature, we employ an unconventional approach to analyze the effect of QE on stock prices, using the DJIA, S&P500, NASDAQ, FTSE100 and the NIKKEI225 as the dependent variable and the balance sheet of the respective Central bank as an explanatory variable along with the unobserved components that account for the behavior of the other explanatory variables that are not explicitly specified in the model. The empirical results show that QE had a significant but not exclusive effect on the DJIA, S&P500 and the NASDAQ, suggesting that these stock prices are also affected by other missing variables and cyclical movements. However, for the UK and Japan, no effect of QE on the FTSE100 and the NIKKEI225 is found, suggesting that variables other than QE are important for the rise in these stock prices.

The remainder of this paper is organized as follows. In the next section, a brief review of the literature on the effect of QE on stock prices is provided, while in Section 3, a description of the structural time series model that is estimated is provided. In Section 4, a description of the data and some preliminary analysis is provided, while Section 5 presents the empirical results. Section 6 provides some concluding remarks.

## 2. The Effect of QE on Stock Prices

The views held by scholars and practitioners about the effect of QE on stock prices are varied. For instance, Galí and Gambetti (2015) use quarterly data for the period 1960–2011 to estimate a six-variable time-varying coefficient structural VAR (TVP-SVAR) for the United States. Since this period also includes the effects of implementing QE, Galí and Gambetti (2015) also estimate their model to the last quarter in 2007 to differentiate the period with QE from that without QE. They highlight that, despite a fall in the short run, share prices remained persistently high, suggesting that QE increased US stock prices. Baumeister and Benati (2013) also estimate a four-variable TVP-SVAR for the United States and the United Kingdom to analyze the effects of a compression in the long-term yield spread using a sample of quarterly observations that ends in 2011. They find that compressions in the long-term yield spread exert a powerful effect on both output growth and inflation, and that QE has averted significant deflation and output collapses similar to those that

occurred during the Great Depression. Wright (2012) attempts to measure the effect of monetary policy surprises on financial variables using daily data from 3 November 2008, to 30 September 2011, by estimating a SVAR during the Subprime Crisis for the United States. Among his many findings, Wright (2012) shows that QE has had a significant effect on financial variables. Florackis et al. (2014) show that there is strong correlation between a lack of market liquidity and the fall in stock prices, using quarterly data over the period 1989–2012 for the United Kingdom. They point out that this occurs during times of low liquidity, such as the Subprime Crisis. One of their main findings is that QE had a positive effect on the economy, boosting economic growth and stock prices. Joyce et al. (2011a, 2011b) indicate that QE may have raised stock prices in the United Kingdom by 20 percent.

Other market participants have made observational inferences about the effect of QE on stock prices. For instance, Newman (2012) states that there is an 'unmistakable correlation' between the Federal Reserve's QE program and the rally in the stock market, while Hubble (2013) points out that when the 'Federal Reserve's balance sheet expands, the stock market rises'. He believes that stock prices have soared since the Federal Reserve launched 'QE Infinity at a pace of $85 billion a month'. Similarly, Lenzner (2014) points to an equal and proportional change between the Federal reserve's purchase of securities and stock prices. He attributes this phenomenal performance to the Fed purchasing $85 billion of Treasury securities and mortgage-backed bonds every month, which has reduced interest rates but increased both bond and stock prices. As a result, Lenzner (2014) concludes that the only factor leading to stock market gains is QE.

Some scholars and market participants, however, disagree with this casual observation. Dobbs et al. (2013) claim that very little evidence is available to support the proposition that QE has caused stock prices to soar, suggesting that the 'conventional wisdom about the effects of QE on equities is probably wrong'. Building on the work of Wallace (1981), Eggertsson and Woodford (2003) formalize the conditions under which QE would be ineffective by developing a model where a single representative agent who has an infinite horizon, faces no credit restrictions and is rational, sees the assets held by the government and by the central bank as indistinguishable from their own assets. In that scenario, any swap of assets with the central bank cannot change anything, an outcome that is analogous to the Ricardian Equivalence hypothesis. However, this result is underpinned by a number of strong assumptions, including that of a representative agent and the assumption of perfect substitutability between assets. As is the case with the Ricardian Equivalence hypothesis, the result obtained by Eggertsson and Woodford (2003) does not hold in more general cases where there are many agents with different preferences, credit constraints, limited financial market participation or distortionary taxes. For instance, Andrés et al. (2004) develops a model with limited participation in financial markets embedded in a DSGE model with agents who have different preferences for government bonds. In that model, purchases by the central bank matter. Similarly, Cúrdia and Woodford (2011) consider the impact of credit imperfections and heterogeneity in a DSGE model that shows that certain types of asset purchases by the central bank can affect demand and output. However, this result only holds when the central bank lends directly to the private sector (credit easing). In this model, QE is ineffective despite Cúrdia and Woodford (2011) eliminating the assumption of a representative agent so that financial intermediation matters, the inclusion of imperfections in private financial intermediation, and the possibility of disruptions to the efficiency of financial intermediation through banks. The reason why QE is ineffective is that Cúrdia and Woodford (2011) think of government bonds as one period claims paying a safe rate identical to the rate set by the central banks, and which, optimally, is the same rate paid on bank reserves. The result is that reserves, or 'money' and government bonds become perfect substitutes. Swapping reserves for QE does nothing.

An important question for which there are several answers is how the effect of asset purchases is transmitted to stock prices. QE is essentially a form of expansionary monetary policy, which implies that the link between the purchases of securities and stock prices

is the same as the connection between the money supply and stock prices. The result of Cúrdia and Woodford (2011) is unique and depends on the fact that the government securities which the central bank buys are short-lived assets with identical characteristics to bank reserves. To generate an impact from QE, investors must not be indifferent to the securities used in readjusting their portfolios. This is why many have viewed the most natural channel through which QE can work is the 'portfolio balance' channel. The nature of this mechanism was initially described by, among others, Tobin (1961, 1963, 1969), Homa and Jaffe (1971), Hamburger and Kochin (1972) and Brunner and Meltzer (1973). They stress how central banks, through varying the relative supplies of financial claims with different maturities and liquidity, could influence the pattern of yields on different assets due to imperfect asset substitutability. This means that the quantities that a central bank could influence, such as the relative quantities of money and government securities held by the private sector, could affect asset prices, including stock prices, and so in turn affect real investment decisions. An important requirement of this portfolio balance channel is heterogeneity across agents.

Even if QE has not led to a massive increase in monetary aggregates (although it has), it has led to ultra-low interest rates, which typically raises stock prices. Furthermore, lower interest rates mean lower borrowing costs, which supports greater productivity and consequently, higher stock prices (See, for instance, Kapetanios et al. (2012) and Chen et al. (2012)). Low interest rates also make it possible for corporations and business firms to borrow money cheaply in order to spend on stock-boosting strategies, such as paying a higher stock dividend or buying back own stock. It is important to note, however, that increases in stock prices during periods of rising interest rates does not rule out the postulated negative relationship between stock prices and interest rates, but rather, it suggests that other factors must have pushed stock prices upwards, dominating the effect of interest rates. Gagnon et al. (2011) describes this channel.

The effect of QE may further work through inflation and economic activity. The underlying premise is that QE stimulates economic activity but is also inflationary. However, the effect of inflation on stock prices is ambiguous. For instance, Lintner (1975, 1978), Fama and Schwert (1977), Feldstein (1980), Fama (1981), Geske and Roll (1983), and Erb et al. (1995) provide evidence of a negative effect, while Firth (1979) and Kaul (1987) find a positive effect. Moreover, Chen et al. (1986) find that inflation has no effect on stock prices. In terms of economic activity, it is plausible to suggest that a thriving economy produces a thriving stock market. Bosworth (1975), Pearce (1983) and Chen et al. (1986) report evidence that is supportive of the positive relationship between real output and stock prices.

Krishnamurthy and Vissing-Jorgensen (2011) and Joyce et al. (2011a, 2011b) suggest that QE distorts financial prices because it involves a manipulation of price signals, which takes the form of lower interest rates, higher demand for assets and a lower purchasing power of money. Instead of stock prices acting as an accurate reflection of company valuation and investor demand, manipulated prices force market participants to adjust their strategies to chase stocks that grow without their underlying companies actually being more valuable. This is yet another channel through which QE affects stock prices.

In more recent times, other scholars and practitioners have analyzed the effects of QE by solely focusing on Treasury yields. For instance, D'Amico and Seida (2020) analyze the effect of quantitative easing and tightening on yields by extracting the unexpected component in balance sheet announcements to assess the sensitivity of Treasury yields for the period of 2009 to 2019. They find that yields do not fall monotonically over time because of quantitative easing and tightening surprises, suggesting that later announcements remain powerful over time. Additionally, D'Amico and Seida (2020) find that yields respond asymmetrically to quantitative easing and tightening surprises, with the effect of surprises in quantitative tightening announcements having a larger effect than that of surprises in quantitative easing announcements. Furthermore, D'Amico and Seida (2020) find that the response of yields is magnified by the amount of uncertainty surrounding interest rate announcements. In a related study, Lucca and Wright (2022) study the effects of QE

policies on yield curve control (where the focus of QE is on targeting a particular yield) using Australian Treasuries, by fitting an affine term structure model and smoothed yield curves to daily data on Australian Treasuries. They find that yield curve control can achieve targets on the selected security, but the effects are narrower with limited spillovers onto the prices of other financial instruments. Additionally, Lucca and Wright (2022) find that liquidity effects dominate preferred habitat and portfolio balance channels, as well the signaling effects about the future path of short-term interest rates. Taken holistically, these findings by Lucca and Wright (2022) suggest that QE policies ease financial conditions but in a narrower way than previously thought. Fabo et al. (2021) undertake a novel approach by examining why the research undertaken by central bankers generally finds a larger and statistically significant effect of QE on inflation and output compared to the research published by academic scholars. Interestingly, they find central bankers whose study report a larger effect of QE on output experience more favorable career outcomes, suggesting that a central banker may be worried that their findings could threaten their employment status and rank.

### 3. Modelling the Relationship between Stock Prices and Quantitative Easing

To analyze the effect of quantitative easing on stock prices, Harvey's (1985, 1989) structural time series model is used. This model is highly appropriate when dealing with a variety of missing variables that may have affected stock prices without the need to specify them, which is particularly poignant in light of the argument proposed by practitioners that quantitative easing is not solely responsible for the increase in stock prices (See, for instance, Dobbs et al. (2013) and Ritholtz (2013)). Another advantage of using this approach is that the model is estimated in a time-varying parametric framework, which is more appropriate than using other estimators that assume parameter estimates do not vary over time.

A structural time series model that relates stock prices to a central bank's balance sheet can be specified as:

$$P_t = \mu_t + \varphi_t + \alpha_t B_t + \varepsilon_t \tag{1}$$

where $P_t$ refers to the respective stock price index and $B_t$ is the balance sheet of the respective central bank. If quantitative easing has any effect on stock prices, the parameter, $\alpha_t$, should be statistically significant. The terms $\mu_t$, $\varphi_t$ and $\varepsilon_t$ represent the structural time series components of the respective stock prices, where $\mu_t$ is the trend component, $\varphi_t$ is the cyclical component and $\varepsilon_t$ is the stochastic component.

The trend component is given by:

$$\mu_t = \mu_{t-1} + \beta_{t-1} + \eta_t \tag{2}$$

$$\beta_t = \beta_{t-1} + \zeta_t \tag{3}$$

where $\eta_t \sim N\left(0, \sigma_\eta^2\right)$ and $\zeta_t \sim N\left(0, \sigma_\zeta^2\right)$. Koopman et al. (2000) show that Equations (2) and (3) specify the trend as a random walk with a drift factor, $\beta_t$, which itself follows a first-order autoregressive process. The trend follows a simple random walk with drift if $\sigma_\zeta^2 = 0$, and a deterministic linear path if $\sigma_\eta^2 = 0$. If, on the other hand, $\sigma_\eta^2 = 0$ and $\sigma_\zeta^2 \neq 0$, then the process will follow a smooth trend.

The cyclical component, which is assumed to be a stationary linear process, may be represented by:

$$\varphi_t = a cos\theta t + b sin\theta t \tag{4}$$

where $t$ is time, and the amplitude of the cycle is given by $\left(a^2 + b^2\right)^{1/2}$. To make the cycle stochastic, the parameters $a$ and $b$ are allowed to evolve over time, while continuity is preserved by writing down a recursion for constructing $\varphi_t$ before introducing the stochastic components. By introducing disturbances and a damping factor, we obtain:

$$\varphi_t = \rho\left(\varphi_{t-1} cos\theta + \varphi_{t-1}^* sin\theta\right) + \omega_t \tag{5}$$

$$\varphi_t^* = \rho\left(-\varphi_{t-1}sin\theta + \varphi_{t-1}^*cos\theta\right) + \omega_t^* \tag{6}$$

where $\varphi^*$ appears by construction, such that $\omega_t$ and $\omega_t^*$ are uncorrelated white noise disturbances with variances $\sigma_\omega^2$ and $\sigma_{\omega^*}^2$, respectively. The parameters $0 \leq \theta \leq \pi$ and $0 \leq \rho \leq 1$ are the frequency of the cycle and the damping factor on the amplitude, respectively, while the period, which is the time taken for the cycle to complete its sequence of values, is given by $2\pi/\theta$.

The extent to which the trend and cyclical components in stock prices evolve over time depends on the values of $\sigma_\eta^2, \sigma_\zeta^2, \sigma_\omega^2, \theta$, and $\rho$, which are known as hyperparameters. These hyperparameters, along with the components, can be estimated by maximum likelihood using the Kalman filter to update the state vector. Importantly, if the coefficient on the central bank's balance sheet, $\alpha_t$, is statistically significant while the trend and/or cyclical components are also significant, then that suggests that while the explanatory variable is an important determinant of stock prices, there are other important variables that affect stock prices whose impact are captured in the trend and cycle. On the other hand, if only the coefficient on the central bank's balance sheet is significant, then this suggests that only the independent variable determines stock prices. Finally, if the trend and/or cyclical components are significant, but the explanatory variable is insignificant, then this suggests that stock prices are determined by variables other than the central bank's balance sheet.

## 4. The Data and Preliminary Analysis

In order to analyze the effect of QE on US, UK and Japanese stock prices, we use monthly observations covering the period that extends between November 2008 and May 2017, with differences for each country. Data on the balance sheet of the Federal Reserve and the Bank of Japan were extracted from the FRED database compiled by the Federal Reserve Bank of St Louis, while data on the balance sheet of the Bank of England was obtained from its *Bank of England Weekly Report 8* (various issues). Monthly observations on the Dow Jones Industrial Average (DJIA), Standard and Poor's 500 (S&P500), National Association of Securities Dealers Automated Quotations (NASDAQ), Financial Times Stock Exchange 100 (FTSE100) and the NIKKEI 225 (NIKKEI225) were obtained from Yahoo! Finance. All variables are measured in natural logarithms for the purposes of estimation.

The data is divided up to encompass the successive rounds of QE that each central bank has engaged in, as well as for the entire period. For the US, the entire period for which the Federal Reserve engaged in QE is 2008:11 to 2014:10. This is divided up into three subperiods: QE1 (2008:11 to 2010:03), QE2 (2010:11 to 2012:06) and QE3 (2012:09 to 2014:10). The Bank of England engaged in QE for the period of 2009:03 to 2017:05. This is divided up into four subperiods corresponding to the four rounds that the Bank of England engaged in QE: QE1 (2009:03 to 2010:01), QE2 (2011:10 to 2012:05), QE3 (2012:07 to 2012:11), and QE4 (2016:08 to 2017:05). The Bank of Japan engaged in QE over the period of 2010:10 to 2016:06. This is divided into two subperiods, that being QE1 (2010:10 to 2013:03) and QE2 (2013:03 to 2016:06), in which the Bank of Japan engaged in what it referred to as 'qualitative and quantitative easing'.[1]

Table 1 shows the percentage changes in the respective central bank's balance sheet and stock prices for the different rounds of QE as well as the whole period. For the Fed, the highest positive association between its balance sheet and the DJIA and S&P500 occurs during QE3. Over the entire period of QE, the Fed's balance sheet increased by almost 110%, while the NASDAQ increased by 135%, the most by any stock price index. Figure 1 shows a series of scatter diagrams that illustrate the relationship between US stock prices and the Fed's balance sheet over the entire period of QE as well as the subperiods of QE1, QE2 and QE3. For the Bank of England, its balance sheet more than doubled over the entire period of QE while the FTSE100 increased by almost 85%, with the highest positive correlation occurring during QE3. Figure 2 presents the scatter diagrams that show the relationship between the FTSE100 and the balance sheet for the Bank of England. The highest positive association between the balance sheet of the Bank of Japan and the NIKKEI 225 occurs during QE2, with its balance sheet more than doubling over the entire period and stock

prices rising by 70%. Figure 3 presents the scatter diagrams that show the relationship between the NIKKEI 225 and the balance sheet for the Bank of Japan. While these initial results provide some indication of the effect of QE on stock prices, the role of other factors should not be dismissed, as suggested by Olsen (2014).

**Table 1.** Changes in Balance Sheets of Central Banks and Stock Prices.

| Operation: | From: | To: | % Δ in BS | % Δ in DJIA | % Δ in S&P500 | % Δ in NASDAQ | % Δ in FTSE100 | % Δ in NIKKEI225 |
|---|---|---|---|---|---|---|---|---|
| **The Federal Reserve** | | | | | | | | |
| QE1 | 2008:11 | 2010:03 | 12.52 | 22.97 | 30.16 | 60.18 | | |
| $\rho_{x,y}$ (*t-statistic*) | | | | 0.808 (5.317) *** | 0.739 (4.253) *** | 0.645 (3.268) *** | | |
| QE2 | 2010:11 | 2012:06 | 23.52 | 13.98 | 11.74 | 18.69 | | |
| $\rho_{x,y}$ (*t-statistic*) | | | | 0.441 (2.082) * | 0.218 (0.946) | 0.483 (2.340) ** | | |
| QE3 | 2012:09 | 2014:10 | 58.62 | 24.47 | 37.79 | 40.54 | | |
| $\rho_{x,y}$ (*t-statistic*) | | | | 0.979 (23.564) *** | 0.988 (31.481) *** | 0.974 (20.862) *** | | |
| QE | 2008:11 | 2014:10 | 108.37 | 51.98 | 60.75 | 135.26 | | |
| $\rho_{x,y}$ (*t-statistic*) | | | | 0.953 (26.254) *** | 0.967 (31.950) *** | 0.966 (31.043) *** | | |
| **Bank of England** | | | | | | | | |
| QE1 | 2009:03 | 2010:01 | 37.99 | | | | 32.15 | |
| $\rho_{x,y}$ (*t-statistic*) | | | | | | | 0.821 (4.307) *** | |
| QE2 | 2011:10 | 2012:05 | 43.15 | | | | −4.03 | |
| $\rho_{x,y}$ (*t-statistic*) | | | | | | | 0.069 (0.167) | |
| QE3 | 2012:07 | 2012:11 | 13.47 | | | | 4.11 | |
| $\rho_{x,y}$ (*t-statistic*) | | | | | | | 0.966 (6.520) *** | |
| QE4 | 2016:08 | 2017:05 | 26.43 | | | | 10.89 | |
| $\rho_{x,y}$ (*t-statistic*) | | | | | | | 0.898 (5.794) *** | |
| QE | 2009:03 | 2017:05 | 219.05 | | | | 84.20 | |
| $\rho_{x,y}$ (*t-statistic*) | | | | | | | 0.847 (15.669) *** | |
| **Bank of Japan** | | | | | | | | |
| QE1 | 2010:10 | 2013:03 | 35.74 | | | | | 29.50 |
| $\rho_{x,y}$ (*t-statistic*) | | | | | | | | 0.229 (1.243) |
| QE2 | 2013:03 | 2016:06 | 162.58 | | | | | 31.24 |
| $\rho_{x,y}$ (*t-statistic*) | | | | | | | | 0.737 (6.722) *** |
| QE | 2010:10 | 2016:06 | 256.42 | | | | | 69.95 |
| $\rho_{x,y}$ (*t-statistic*) | | | | | | | | 0.900 (16.857) *** |

*Notes*: The term $\rho_{x,y}$ refers to the correlation between the respective central bank's balance sheet and the stock price, while the numbers reported in parenthesis are the t-*statistics* associated with that correlation coefficient. An *, **, and *** indicates that the null hypothesis of no statistical significance is rejected at the 10, 5 and 1 percent levels, respectively.

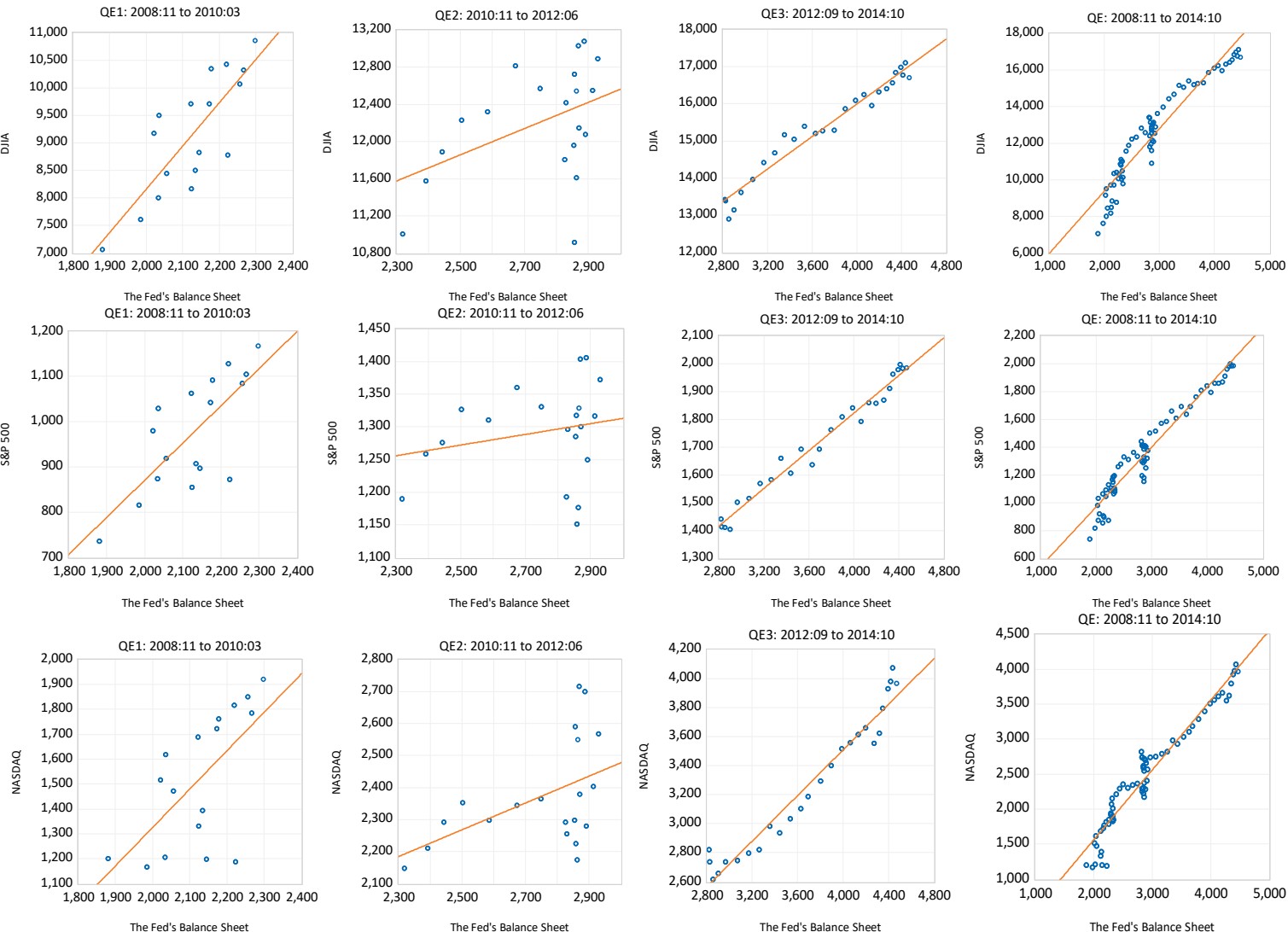

**Figure 1.** Scatter plot of stock prices against the Federal Reserve's balance sheet (in billions).

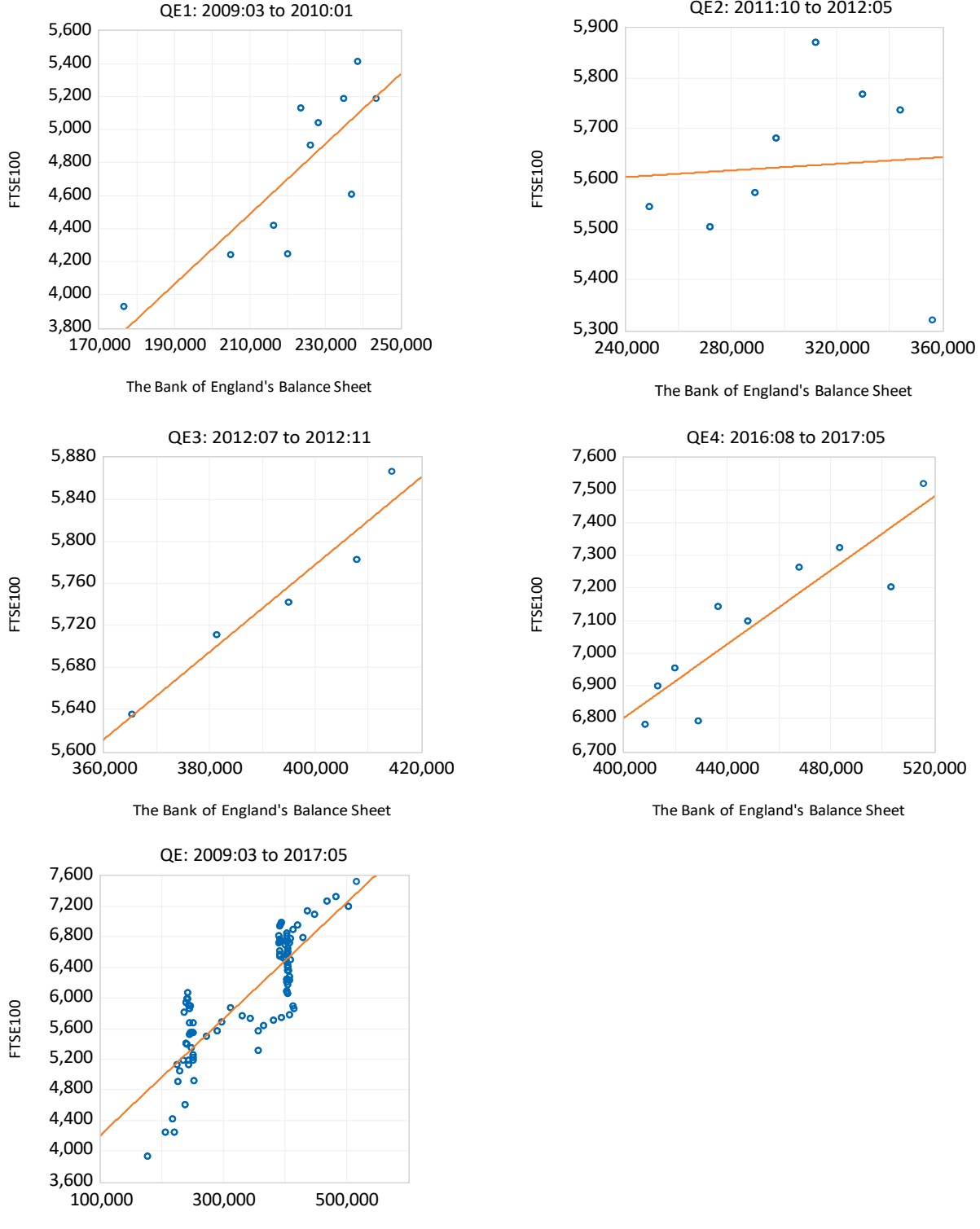

**Figure 2.** Scatter plot of the FTSE100 against the Bank of England's balance sheet (in millions).

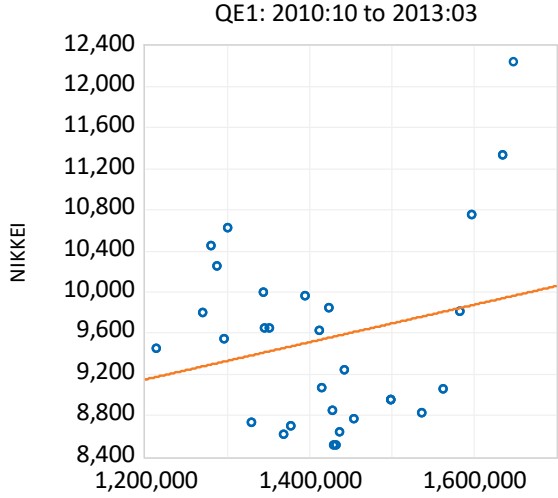

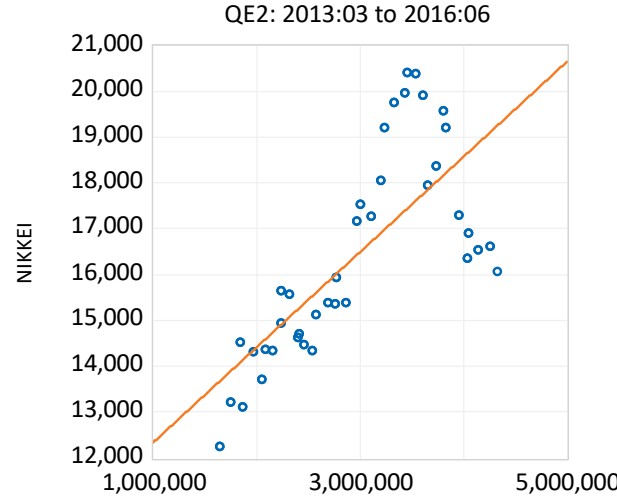

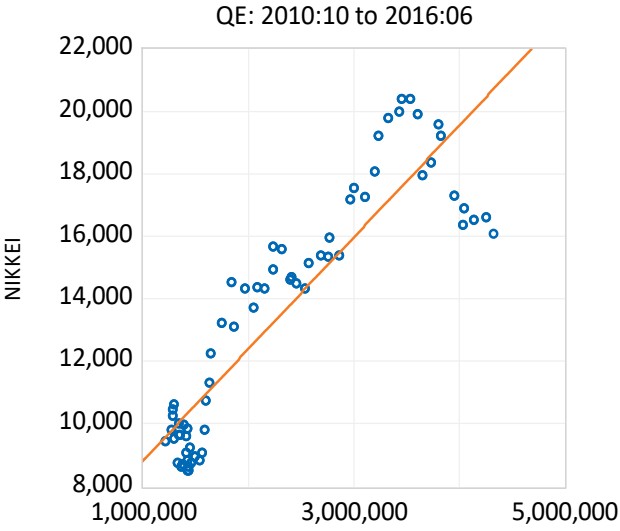

**Figure 3.** Scatter plot of the NIKKEI 225 against the Bank of Japan's balance sheet (in millions).

## 5. Empirical Results

The empirical results of analyzing the effect of QE on stock prices for the US are presented in Table 2, which shows the results of estimating the model for the three rounds of QE undertaken by the Fed, QE1 (2008:11 to 2010:03), QE2 (2010:11 to 2012:06) and QE3 (2012:09 to 2014:10), as well as for the entire period, QE (2008:11 to 2014:10). Table 2 also reports the estimated components and coefficients of the final state vector, as well as some measures for the overall goodness of fit, such as the Akaike Information Criterion, $AIC$, Bayesian Information Criterion, $BIC$, and the coefficient of determination, $R^2$. It also reports the diagnostic tests for heterscedasticity, serial correlation and normality. The $H$ test is a test statistic for heteroscedasticity and has an $F$ distribution, while the $DW$ test is the Durbin and Watson (1971) test and the $Q$ test is the Ljung and Box (1978) test for serial correlation, which has a $\chi^2$ distribution. Finally, $N$ is the Bowman and Shenton (1975) test for normality.

**Table 2.** Results of Estimating the Model for the Federal Reserve.

| Component | QE1 | QE2 | QE3 | QE |
|---|---|---|---|---|
| | **Dow Jones Industrial Average** | | | |
| $\mu_t$ | −4.443 | 7.061 *** | −0.655 | −1.716 |
| | (−1.495) | (4.021) | (−0.248) | (−0.695) |
| $\beta_t$ | 0.014 * | 0.002 | −0.004 | $3.60 \times 10^{-4}$ |
| | (1.876) | (1.200) | (−1.046) | (0.079) |
| $\varphi_t$ | −0.042 ** | 0.007 | 0.005 | $8.79 \times 10^{-3}$ |
| | (−2.460) | (1.080) | (0.410) | (0.440) |
| $\varphi_t^*$ | 0.028 | −0.058 *** | −0.011 | $−2.53 \times 10^{-3}$ |
| | (1.657) | (−8.738) | (−0.909) | (−0.125) |
| $\alpha_t$ | 0.940 *** | 0.159 | 0.678 *** | 0.747 *** |
| | (4.644) | (1.350) | (3.948) | (4.632) |
| $AIC$ | −6.463 | −7.700 | −8.346 | −6.732 |
| $BIC$ | −6.267 | −7.501 | −8.152 | −6.605 |
| $DW$ | 1.850 | 2.167 | 1.873 | 1.968 |
| $R^2$ | 0.9430 | 0.8962 | 0.9787 | 0.9787 |
| $H$ | 0.741 | 0.100 | 0.490 | 0.168 |
| $Q$ | 12.02 *** | 2.342 | 6.271 ** | 14.58 ** |
| $N$ | 0.361 | 19.374 *** | 0.217 | 2.102 |
| | **S&P500** | | | |
| $\mu_t$ | −4.619 | −8.960 ** | −1.171 | −0.959 |
| | (−1.739) | (−2.130) | (−0.453) | (−0.368) |
| $\beta_t$ | 0.020 *** | −0.008 ** | 0.003 | 0.006 * |
| | (10.002) | (−2.420) | (0.884) | (1.710) |
| $\varphi_t$ | −0.042 *** | 0.069 *** | −0.001 | 0.001 |
| | (−4.057) | (6.824) | (−0.088) | (0.068) |
| $\varphi_t^*$ | 0.035 *** | −0.061 *** | $−9.00 \times 10^{-5}$ | $−6.00 \times 10^{-5}$ |
| | (3.102) | (−4.104) | (−0.007) | (−0.003) |
| $\alpha_t$ | 0.800 *** | 1.081 *** | 0.573 *** | 0.559 *** |
| | (4.405) | (3.838) | (3.407) | (3.283) |
| $AIC$ | −6.678 | −6.887 | −7.759 | −6.401 |
| $BIC$ | −6.482 | −6.688 | −7.565 | −6.274 |
| $DW$ | 1.675 | 2.253 | 2.122 | 1.833 |
| $R^2$ | 0.9600 | 0.8090 | 0.9793 | 0.9761 |
| $H$ | 0.391 | 1.252 | 0.208 | 0.280 |
| $Q$ | 2.325 | 4.544 ** | 5.441 ** | 18.29 *** |
| $N$ | 1.190 | 15.198 *** | 0.865 | 0.559 |

**Table 2.** *Cont.*

| Component | QE1 | QE2 | QE3 | QE |
|---|---|---|---|---|
| | | NASDAQ | | |
| $\mu_t$ | 3.148 | 8.315 | −8.334 | 5.971 ** |
| | (1.458) | (1.070) | (−0.941) | (2.220) |
| $\beta_t$ | 0.032 *** | 0.009 | 0.002 | 0.015 *** |
| | (22.136) | (1.283) | (0.199) | (3.490) |
| $\varphi_t$ | −0.050 *** | −0.001 | −0.004 | −0.012 |
| | (−6.331) | (−0.018) | (−0.178) | (−0.766) |
| $\varphi_t^*$ | 0.002 *** | −0.032 | −0.004 | 0.026 |
| | (0.194) | (−0.636) | (−0.156) | (1.629) |
| $\alpha_t$ | 0.304 * | −0.032 | 1.086 * | 0.152 |
| | (2.057) | (−0.061) | (1.878) | (0.866) |
| AIC | −7.255 | −6.306 | −7.205 | −6.674 |
| BIC | −7.059 | −6.107 | −7.012 | −6.548 |
| DW | 2.370 | 2.361 | 1.641 | 1.990 |
| $R^2$ | 0.9882 | 0.7674 | 0.9756 | 0.9898 |
| H | 0.408 | 1.220 | 1.217 | 0.323 |
| Q | 2.373 | 1.868 | 2.614 | 16.747 *** |
| N | 0.194 | 0.994 | 1.110 | 4.485 |

*Notes*: The full period for which the Federal Reserve undertook QE is divided up into three subperiods, these being QE1 (2008:11 to 2010:03), QE2 (2010:11 to 2012:06) and QE3 (2012:09 to 2014:10), while the full time period for QE is 2008:11 to 2014:10. The *H*, *Q* and *N* statistics are distributed as $F(4,4)$, $\chi^2(1)$, and $\chi^2(2)$ for QE1, $F(5,5)$, $\chi^2(1)$, and $\chi^2(2)$ for QE2, $F(7,7)$, $\chi^2(1)$, and $\chi^2(2)$ for QE3, and $F(23,23)$, $\chi^2(5)$, and $\chi^2(2)$ for the entire QE period. The figures in parenthesis are the *t*-statistics. An *, **, and *** indicates that the null hypothesis of no statistical significance is rejected at the 10, 5 and 1 percent levels, respectively.

The results of estimating the effect of QE on the DJIA for the QE1 period shows that the equation is well determined in terms of the goodness of fit and passes the diagnostic tests for heteroscedasticity and normality but fails the test for serial correlation. The slope of the trend is significant and so is the cycle. Furthermore, the significance of the coefficient on the explanatory variable suggests that QE has had a positive effect on the DJIA over that period. However, the significance of the slope of the trend and the cycle shows that there are some missing variables not included in the model that affect the secular and cyclical movements of the DJIA. For QE2, the level of the trend as well as the cyclical component are significant, implying that there are important missing determinants which affect the secular and cyclical movements of the DJIA over this time period. However, the coefficient on the explanatory variable is insignificant during this round of QE, suggesting that QE had no effect of the DJIA over the period, 2010:11 to 2012:06. For QE3, the only significant variable is the Fed's balance sheet, suggesting that QE was the only determinant of the DJIA over the period of 2012:09 to 2014:10. Moreover, for the entire period that the Fed undertook QE, 2008:11 to 2014:10, the only variable that is significant is the Fed's balance sheet, suggesting that for the entire period under analysis, the only determinant of the DJIA is QE.

Similar findings are obtained for the effect of QE on the S&P500. For the first round of QE, the results show that the slope of the trend and the cyclical component are significant at the 1 per cent level. Additionally, the Fed's balance sheet is also statistically significant at the 1 per cent level, suggesting that not only are there missing variables that affect the secular and cyclical movements of the S&P500, but QE is also a significant determinant of the S&P500 over the period of 2008:11 to 2010:03. For the second round of QE, the

results show that the level and slope of the trend as well as the cyclical component of the S&P500 are statistically significant. They also show that QE is a significant determinant of the S&P500 over this round of QE. However, the estimated model fails the test for serial correlation and normality. For the third round of QE, the estimated model passes the test for heteroscedasticity and normality and is very well determined, with the only significant variable being the Fed's balance sheet, suggesting that for this period of QE, the only determinant of the S&P500 is QE. Over the entire period of QE, the only structural time series component that is significant is the slope of the trend, along with the explanatory variable, suggesting that QE is an important determinant of S&P500.

The results obtained for the NASDAQ are similar to those obtained for the DJIA and S&P500. For the first round of QE, the results show that model is very well specified and passes every diagnostic test. They also show that the slope of the trend and the cyclical component are statistically significant, as well as the Fed's balance sheet. This suggests that not only is QE an important determinant of the NASDAQ during this round of QE, but there are also some important determining variables that are missing from the model. For the second round of QE, no structural time series component or explanatory variable is significant, while for the third round of QE, the only significant variable is the Fed's balance sheet. This suggests that over the period of 2012:09 to 2014:10, QE was the only determinant of the NASDAQ. When the model is estimated for the entire period of QE, the level and the slope of the trend are the only significant components, suggesting missing important variables that affect the secular movements of the NASDAQ. These results for the US support the proposition put forward by Lenzner (2014) that QE worked reasonably well to propel stock prices.

The empirical results of analyzing the effect of QE on stock prices for the UK are presented in Table 3.[2] During the first round of QE that the Bank of England engaged in, the level of the trend and the cyclical component are statistically significant, suggesting that important missing variables affect the secular and cyclical movements of the FTSE100. The remaining results show very few significant components and coefficients, with the Bank of England's balance sheet being statistically insignificant across all periods that it undertook QE in. Similar results are obtained for the effect of QE undertaken by the Bank of Japan on the NIKKEI 225, which are presented in Table 4. For the first period that QE was engaged in, from 2010:10 to 2013:03, the results show that the level and slope of the trend of the NIKKEI 225 are statistically significant, but the coefficient on the explanatory variable is not, suggesting that important variables are excluded from the model. The results obtained for the UK and Japan show that QE had no effect whatsoever on the FTSE100 and the NIKKEI 225, respectively.

Taken together as a whole, these results suggest that it is plausible to assume that portfolio managers shift between equity and fixed-income securities, which implies that the effect of QE through portfolio adjustment does work, unlike the assertion made by Dobbs et al. (2013). Additionally, the effect of QE is transmitted through lower interest rates, which affects stock prices through many channels, as outlined earlier. Some of the missing variables that are considered to be important determinants of stock prices have been outlined by, for instance, Olsen (2014) who suggests that earnings and growing dividends are better explanatory variables of stock prices than QE. Similarly, Blanchard et al. (2018), suggest that some of the important missing variables may include actual and expected dividends, the decline in the equity premium and lower uncertainty in the rest of the world.

The results obtained from estimating the model suggests that while QE generally had a significant effect on the DJIA, S&P500 and the NASDAQ, it had no effect on the FTSE100 and the NIKKEI 225, suggesting that it is not the only determining factor. This finding somewhat supports Lenzner's (2014) proposition that QE worked reasonably well to propel US stock prices but not for UK or Japanese stock prices.[3] On the other hand, these results also support Ritholtz's (2013) contention that the performance of the US stock market cannot be exclusively attributed to QE.

**Table 3.** Results of Estimating the Model for the Bank of England.

| Component | QE1 | QE2 | QE3 | QE4 | QE |
|---|---|---|---|---|---|
| $\mu_t$ | 6.720 ** | 10.942 | NA | 9.324 | 6.808 |
| | (2.529) | (1.224) | NA | (1.544) | (3.954) |
| $\beta_t$ | 0.021 | −0.058 | NA | 0.012 | 0.005 |
| | (1.754) | (−1.366) | NA | (0.803) | (1.255) |
| $\varphi_t$ | −0.007 | 0.000 | NA | −0.000 | −0.003 |
| | (−0.771) | (0.005) | NA | (−0.003) | (−0.267) |
| $\varphi_t^*$ | −0.024 ** | −0.000 | NA | 0.000 | 0.003 |
| | (−2.643) | (−0.002) | NA | (0.001) | (0.276) |
| $\alpha_t$ | 0.148 | −0.185 | NA | −0.031 | 0.161 |
| | (0.694) | (−0.264) | NA | (−0.068) | (1.228) |
| AIC | −6.533 | −5.986 | NA | −7.149 | −6.334 |
| BIC | −6.388 | −5.946 | NA | −7.028 | −6.529 |
| DW | 1.614 | 1.490 | NA | 2.310 | 1.994 |
| $R^2$ | 0.9493 | 0.3217 | NA | 0.7262 | 0.9283 |
| H | 0.521 | 285.150 ** | NA | 2.136 ** | 0.434 |
| Q | 4.263 ** | 1.384 | NA | 9.553 *** | 10.585 |
| N | 0.858 | 0.604 | NA | 0.091 | 4.996 |

*Notes*: The full period for which the Bank of England undertook QE is divided up into four subperiods, these being QE1 (2009:03 to 2010:01), QE2 (2011:10 to 2012:05), QE3 (2012:07 to 2012:11), QE4 (2016:08 to 2017:05) while the full time period for QE is 2009:03 to 2017:05. The $H$, $Q$ and $N$ statistics are distributed as $F(2,2)$, $\chi^2(1)$, and $\chi^2(2)$ for QE1, $F(1,1)$, $\chi^2(3)$, and $\chi^2(2)$ for QE2, $F(23,23)$, $\chi^2(1)$, and $\chi^2(2)$ for QE4, $F(32,32)$, $\chi^2(19)$, and $\chi^2(2)$ for the entire QE period. The figures in parenthesis are the *t*-statistics. An *, **, and *** indicates that the null hypothesis of no statistical significance is rejected at the 10, 5 and 1 percent levels, respectively.

**Table 4.** Results of Estimating the Model for the Bank of Japan.

| Component | QE1 | QE2 | QE |
|---|---|---|---|
| $\mu_t$ | 13.057 *** | 4.254 | 12.043 *** |
| | (4.306) | (0.464) | (3.336) |
| $\beta_t$ | 0.063 * | −0.010 | 0.005 |
| | (2.000) | (−0.642) | (0.344) |
| $\varphi_t$ | 0.011 | −0.071 *** | −0.093 |
| | (0.186) | (−2.782) | (−0.731) |
| $\varphi_t^*$ | 0.024 | 0.003 | −0.092 |
| | (0.416) | (0.134) | (−0.745) |
| $\alpha_t$ | −0.255 | 0.361 | −0.148 |
| | (−1.205) | (0.602) | (−0.629) |
| AIC | −5.919 | −5.920 | −5.886 |
| BIC | −5.732 | −5.751 | −5.756 |
| DW | 1.656 | 1.543 | 1.934 |
| $R^2$ | 0.7628 | 0.8842 | 0.9721 |
| H | 0.497 | 1.232 | 1.220 |
| Q | 20.566 | 22.138 | 12.275 |
| N | 1.456 | 3.075 | 2.883 |

*Notes*: The full period for which the Bank of Japan undertook QE is divided up into two subperiods, these being QE1 (2010:10 to 2013:03) and QE2 (2013:03 to 2016:06), while the full time period for QE is 2010:10 to 2016:06. The $H$, $Q$ and $N$ statistics are distributed as $F(9,9)$, $\chi^2(19)$, and $\chi^2(2)$ for QE1, $F(12,12)$, $\chi^2(19)$, and $\chi^2(2)$ for QE2, $F(22,22)$, $\chi^2(19)$, and $\chi^2(2)$ for the entire QE period. The figures in parenthesis are the *t*-statistics. An *, **, and *** indicates that the null hypothesis of no statistical significance is rejected at the 10, 5 and 1 percent levels, respectively.

### 6. Concluding Remarks

The scale of QE can be measured by the size and growth of a Central Bank's balance sheet as it accumulates securities bought in exchange for newly printed money. The stylized facts generally show a strong relationship between stock prices and a Central Bank's balance sheet. However, there is no consensus on the view that QE affects stock prices. The results presented in this paper show that QE undertaken by the Fed had a significant but not exclusive effect on the DJIA, S&P500 and the NASDAQ. The empirical results derived from estimating the structural time series model shows that, in addition to QE, stock prices were affected by missing variables. However, for the FTSE100 and the NIKKEI 225, QE undertaken by the Bank of England and the Bank of Japan respectively, had no effect. One possible explanation for this result is that perhaps QE becomes effective only after a certain threshold level is met, one that the Federal Reserve has met but not the Bank of England nor the Bank of Japan.

**Author Contributions:** Conceptualization, I.A.M.; methodology, I.A.M.; software, G.B.T.; validation, I.A.M. and G.B.T.; formal analysis, G.B.T.; investigation, G.B.T.; resources, G.B.T.; data curation, G.B.T.; writing—original draft preparation, G.B.T.; writing—review and editing, I.A.M. and G.B.T.; visualization, G.B.T.; supervision, I.A.M.; project administration, G.B.T.; funding acquisition, none. All authors have read and agreed to the published version of the manuscript.

**Funding:** This research received no external funding.

**Informed Consent Statement:** Not Applicable.

**Data Availability Statement:** The data presented in this study are available on request from the corresponding author (George B. Tawadros).

**Conflicts of Interest:** The authors declare no conflict of interest.

### Notes

[1]    For more details on the rounds and duration of QE that the Federal Reserve, the Bank of England and the Bank of Japan have implemented, see Farmer (2012), Joyce et al. (2011a, 2011b), Joyce et al. (2012), Joyce and Tong (2012), Kapetanios et al. (2012), Agostini et al. (2016), Greenwood (2017), and Al-Jassar and Moosa (2019).

[2]    A model corresponding to the third round of QE that the Bank of England engaged in could not be estimated because of a lack of observations.

[3]    One plausible explanation for this result is that perhaps QE becomes effective only after a certain threshold level is met.

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
