# Peer review of "A Structural Time Series Analysis of the Effect of Quantitative Easing on Stock Prices"

_ijfs, doi:10.3390/ijfs10040114_

Round 1

Reviewer 1 Report

The introduction is very informative and sets up the main tenet of the paper which is to analyse the effect of quantitative easing (QE) on stock prices for the US, UK and Japan. It also leads naturally to the main objective of the paper.

The literature review provides insights in QE and the debate which surrounds both its use and potential impacts from applying it. Although a good overall review, the authors should consider the inclusion of a few more recent studies into this section.

The research methods section is fine and the choice of a structural time series model to estimate via a time-varying parametric framework is appropriate. The discussion of the results is quite through and linked back to the literature. This section is arguably the main strength of the manuscript.

The conclusion section requires further work. Specifically, the authors need to explicitly address the limitations of the study and provide some avenues for further research. In addition, there is a lack of managerial/policy implications arising from the research. This should also be included.

Author Response

1st Reviewer Report

Comments and Suggestions for Authors

The introduction is very informative and sets up the main tenet of the paper which is to analyse the effect of quantitative easing (QE) on stock prices for the US, UK and Japan. It also leads naturally to the main objective of the paper.

We thank the reviewer for their positive and generous comments about the introduction that we wrote. We have included an additional write-up in the introduction, which is:

“Unlike the extant literature, we employ an unconventional approach to analyze the effect of QE on stock prices, using the DJIA, S&P500, NASDAQ, FTSE100 and the NIKKEI225 as the dependent variable and the balance sheet of the respective Central bank as an explanatory variable along with the unobserved components that account for the behavior of the other explanatory variables that are not explicitly specified in the model. The empirical results show that QE had a significant but not exclusive effect on the DJIA, S&P500 and the NASDAQ, suggesting that these stock prices are also affected by other missing variables and cyclical movements. However, for the UK and Japan, no effect of QE on the FTSE100 and the NIKKEI225 is found, suggesting that variables other than QE are important for the rise in these stock prices.”

The literature review provides insights in QE and the debate which surrounds both its use and potential impacts from applying it. Although a good overall review, the authors should consider the inclusion of a few more recent studies into this section.

We thank the reviewer for this suggestion. Again, we believe that the reviewer may have misunderstood the scope of the paper, which is to narrowly focus on the effect that QE has on stock prices. We do not develop a theoretical New-Keynesian DSGE model and calibrate it because is beyond the scope of the paper. This paper is an empirical paper using an unconventional econometric technique that only one other paper has employed, that being that by Al-Jassar and Moosa (2019). This study builds on Al-Jassar and Moosa’s (2019) paper. Nevertheless, because of the reviewer’s suggestion, we have cited the following studies and included them in the reference list:

D’Amico, S. and Seida, T., (2020), “Unexpected Supply effects of Quantitative Easing and Tightening”, Federal Reserve Bank of Chicago, Working Paper No. 2020-17.

Fabo, B., Jan oková, M., Kempf, E. and Pástor, U., (2021), “Fifty Shades of QE: Comparing Findings of Central Bankers and Academics”, National Bureau of Economic Research, Working Paper No. 27849.

Lucca, D. O. and Wright, J. H, (2022), “The Narrow Channel of Quantitative Easing: Evidence from YCC Down Under”, Federal Reserve Bank of New York, Staff Report No. 1013.

However, its important to note that these studies do not analyse the effect of QE on stock prices, so we believe that they are largely irrelevant to objective of the study. It is important to note that there are generally very few academic papers analyzing the effect of QE on stock prices but a lot of practitioner-based pieces on the internet, which are hardly academic pieces of writing. Nevertheless, we have included them in the paper.

The research methods section is fine and the choice of a structural time series model to estimate via a time-varying parametric framework is appropriate. The discussion of the results is quite through and linked back to the literature. This section is arguably the main strength of the manuscript.

We thank the reviewer for their positive and generous comments about the research methodology section.

The conclusion section requires further work. Specifically, the authors need to explicitly address the limitations of the study and provide some avenues for further research. In addition, there is a lack of managerial/policy implications arising from the research. This should also be included.

We thank the reviewer for this particular suggestion. However, we follow the conventional practice of linking our conclusions to the objective of the study and summarising our findings. We don’t highlight the limitations of our study because no one in their right mind would want to show the deficiencies and flaws of their work, and therefore provide the reviewers with an ability to easily reject their paper outright. We have, however, included the following passage in the conclusion:

One possible explanation for this result is that perhaps QE becomes effective only after a certain threshold level is met, one that the Federal Reserve has met but not the Bank of England nor the Bank of Japan.

Reviewer 2 Report

This paper is interesting and the analyses on this topic are well structured. 

However, there are some points that would improve the paper.

Analyses based on existing studies are not enough. The difference between this study and existing ones is not clear. The author should mention clearly what and where other authors’ ideas are cited that would improve this paper greatly. It is also tant that the paper provide something new that differentiates it from existing studies.

The theoretical background seems to be weak in general. Listed studies are old. Citing comparatively new studies is recommended.

A detailed explanation of the data is required.

Author Response

2nd Reviewer Report

Comments and Suggestions for Authors

This paper is interesting and the analyses on this topic are well structured. 

However, there are some points that would improve the paper.

Analyses based on existing studies are not enough. The difference between this study and existing ones is not clear. The author should mention clearly what and where other authors’ ideas are cited that would improve this paper greatly. It is also tant that the paper provide something new that differentiates it from existing studies.

We thank the reviewer for this suggestion. However, we believe that the reviewer may have misunderstood that the focus of the paper was the effect of QE on stock prices only and not on output, employment and inflation. Nevertheless, we have incorporated the reviewer’s suggestion and included the following piece in the introduction:

“Unlike the extant literature, we employ an unconventional approach to analyze the effect of QE on stock prices, using the DJIA, S&P500, NASDAQ, FTSE100 and the NIKKEI225 as the dependent variable and the balance sheet of the respective Central bank as an explanatory variable along with the unobserved components that account for the behavior of the other explanatory variables that are not explicitly specified in the model. The empirical results show that QE had a significant but not exclusive effect on the DJIA, S&P500 and the NASDAQ, suggesting that these stock prices are also affected by other missing variables and cyclical movements. However, for the UK and Japan, no effect of QE on the FTSE100 and the NIKKEI225 is found, suggesting that variables other than QE are important for the rise in these stock prices.”

The theoretical background seems to be weak in general. Listed studies are old. Citing comparatively new studies is recommended.

We thank the reviewer for this suggestion. Again, we believe that the reviewer may have misunderstood the scope of the paper, which is to narrowly focus on the effect that QE has on stock prices. We do not develop a theoretical New-Keynesian DSGE model and calibrate it because is beyond the scope of the paper. This paper is an empirical paper using an unconventional econometric technique that only one other paper has employed, that being that by Al-Jassar and Moosa (2019). This study builds on Al-Jassar and Moosa’s (2019) paper. Nevertheless, because of the reviewer’s suggestion, we have cited the following studies and included them in the reference list:

D’Amico, S. and Seida, T., (2020), “Unexpected Supply effects of Quantitative Easing and Tightening”, Federal Reserve Bank of Chicago, Working Paper No. 2020-17.

Fabo, B., Jan oková, M., Kempf, E. and Pástor, U., (2021), “Fifty Shades of QE: Comparing Findings of Central Bankers and Academics”, National Bureau of Economic Research, Working Paper No. 27849.

Lucca, D. O. and Wright, J. H, (2022), “The Narrow Channel of Quantitative Easing: Evidence from YCC Down Under”, Federal Reserve Bank of New York, Staff Report No. 1013.

However, its important to note that these studies do not analyse the effect of QE on stock prices, so we believe that they are largely irrelevant to objective of the study. It is important to note that there are generally very few academic papers analyzing the effect of QE on stock prices but a lot of practitioner-based pieces on the internet, which are hardly academic pieces of writing. Nevertheless, we have included them in the paper.

A detailed explanation of the data is required.

We thank the reviewer for this suggestion. However, we believe that the data that is used in this paper has been adequately described and is highly comparable to the other studies that analyze the effect of QE on stock prices.